# Peer review of "A Carbon-Based Antifouling Nano-Biosensing Interface for Label-Free POCT of HbA1c"

_biosensors, 2021, doi:10.3390/bios11040118_

Round 1

Reviewer 1 Report

The paper by Li et al. describes the fabrication of a conductive and antifouling layer to quantify glycated haemoglobin (HbA1c) in solution. The antifouling activity is obtained by crosslinking BSA, while the conductivity in a 3D porous matrix is ensured by using multi-walled carbon nanotubes. Two different strategies for HbA1c detection are proposed, the first based on immobilization by anti- HbA1c antibodies followed by amperometric detection of HbA1c-mediated electroreduction of hydrogen peroxide, the second on binding of HbA1c to APBA. The paper is not exceptionally innovative, as many of the strategies employed have been reported already individually (e.g. crosslinked BSA as antifouling, HbA1c detection by boronic acids, encapsulation of nanomaterials to enhance conductivity) but here they have been merged to demonstrate a sensing architecture that I think might be of interest to the readership of Biosensors. The main issues that should be addressed before publication concern assessment of the reproducibility of the response and an improved description of the electrochemical results and rationale behind the sensing schemes.

Major issues:

1) The authors express HbA1c concentrations as %, which I understand is customary for this biomarker, yet it might create confusion to readers not expert in the field of diabetes. Maybe the authors might want to explain how to interpret HbA1c % levels. Else, it might create some confusion. It is difficult to understand, for example, at line 276 when the authors test the potential interference from non glycated protein describing the solution as composed by “200 μg/ml HbAo and 6% HbA1c 276 in unprocessed human serum”: using two different notations for concentration is misleading and does not allow to grasp the ratio between the concentration of the two proteins. The same happens for the LOD at line 263, expressed in ug/ml and at line 291, expressed as %. Along the same line: what are the units in the legend of Figure 3b?

2) Lines 202-on. Although it is mentioned in Section 2, the authors should remind the reader that the CV signal are obtained for the ferri/ferrocyanide couple in solution. Most importantly, it is not clear whether these experiments have been repeated and, if so, how many times. This would allow estimating the distribution of the observed changes, e.g. the 32% increase of faradaic current (line 211). The same holds for the values extracted from fitting the EIS data with equivalent circuit, where fitting values of Rct are provided with a tenth of Ohm (.1 Ohm) precision that most likely is much lower than the dispersion between different independent samples. How many times were the electrochemical experiments repeated? How reproducible is the response of the modified electrodes? Some information is provided in lines 328 and 329 (is this the origin of the error bars in Figures 3f and 4d?) but more thorough data analysis taking reproducibility into account would be important.

3) Why did the authors express the WE potential vs the open circuit potential rather than reporting it vs a reference electrode? This choice makes it impossible to compare. From scheme 1 it looks like they used a 3-electrode system, what is the reference electrode used?

4) It is not clear to me how was the control experiment with non glycated-protein performed (Figure 3 panel c). Was the same electrode incubated first in HbAo and then in HbA1c? Moreover, what is the difference between the experiment in Figure 3 panels b and d? In my opinion, the whole section describing the electrochemical experiments should be edited to make it clearer to the reader.

5) With respect to the mechanism behind the recognition between APBA and the protein it would be interesting to see if the layer would bind to other glycoprotein (as expected), this might hamper its selectivity for glycated hemoglobin with respect to other glycated proteins. Have control experiments been performed (or have they been reported in the papers describing interaction binding of APBA to glycated proteins)? Within this respect, the reference that is cited for the ability of boronic acid to bind glycated biomolecules, ref. 45, is not correct in my opinion.

Minor issues:

6) Line 57: “rate” should be substituted with “ratio”.

7) The authors use both the abbreviation SPEC and SPCE. They should pick one and use it consistently.

8) Line 173: “through hydrophobic” the authors should rather say “through hydrophobic interaction”

9) Line 240: H2O2, 2 has to be subscript.

10) Line 290: “detection of limit” should be changed into “limit of detection”.

11) There are several other typos, see lines 251, 253, legend of Figure 4a, and many others.

Author Response

Thank you very much for review our manuscript. We are very grateful to see your  comments to help improve the quality of our manuscript. Our point-to-point responses are detailed on the uploaded response, and the changes are highlighted in the revised manuscript. 

Reviewer 2 Report

The manuscript of Li and co-workers describes the design of a three-dimensional electron transporter that improves the efficiency of electron transfer. This new design consists of a layer of BSA and MWCNTs cross-linked with glutaraldehyde. The authors use it as a proof of concept for the detection of HbA1c in plasma. The work is very interesting in the field because the authors detected this human protein using a label-free method.  There are some issues that authors need to clarify:

- The authors use HbA1c standards and in 2.4.3, they explain that they were mixed with unprocessed human plasma, but what about the amount of endogenous HbA1c in this normal donor plasma? The levels of this endogenous HbA1c are determined in the hospital and are subtracted from the standards added by the authors.

-When the authors refer in the results to 0% hemoglobin, do they consider the levels of the endogenous ones?

The Authors will also correct:

-  There are some abbreviations that are cited before the full description of the term, for example, HbA1c and APBA.

- In figure 3 of the legend authors write HbA1o, Is this correct? Or is it supposed to be HbAo.

Author Response

(The authors gave the same response as above.)

Reviewer 3 Report

Line 15: “transporter” is the word for an active transport process. Here the authors only enhance the passive electron transfer. Is there a more suitable expression than “transporter”?

Line 21: “anti-HbA1c”, is it anti-HbA1c antibody? Please explain also the abbreviation “APBA.

Line 45: please explain the causality of “complicated because HbA1c is formed by non-enzymatic glycosylation of hemoglobin”! I don’t see any.

Line 54: graphene instead of “grapheme”

Line 57: ratio instead of “rate”

Line 58: what does this mean “the involvement of inorganic nanomaterials can do nothing for”? It is a bit confusing.

Line 63: Why “decrease of electrode conductivity, decreasing the sensitivity”? See: https://en.wikipedia.org/wiki/Sensitivity_and_specificity

Line 72: “Nevertheless, such kind of developed strategies suffer from two disadvantages. First, the need of noble metal would impede the industrial applications and the spread of these techniques. Second, complicated operations of these methods make them fail to meet the requirement of point-of-care diagnostics.” is a too general statement. First, biosensors are not industrial applications and so noble metals are OK for it. Second, the methods used normally are not more complicate than the methods described by the authors. So please leave out or explain.

Line 77: screen-printed instead of “screen-printing”

Line 91: peroxide (H2O2)

Line 98: China).

Line 104: centrifuged instead of “centrifugated”

Line 108: which electrolyte was used? More information on layer fabrication are necessary.

Line 127: which electrolyte was used?

Line 132: was chosen instead of “was optimized”

Line 132/133: chose instead of “optimized”

Line 134: was chosen instead of “was optimized”

Line 136: which electrolyte was used?

Line 137: what is challenged?

Line 138: 1% BSA in human serum? What was done after 1 week, etc.?

Line 142: which electrolyte was used for NHS, EDC, anti-HbA1c, and APBA?

Line 143: which electrolyte was used?

Line 144: please explain rinsing

Line 145: chapter 2.4.2 experimental details have to be presented more precisely!

Line 148: analysis instead of “detection”

Line 150: analysis instead of “detection”

Line 150: EIS was already explained before in chapter 2.3.3.

Line 170: are the MWCNTs unmodified or modified with whatever? How can MWCNTs cross-link with GA and BSA.

Line 168: how the layer can increase sensitivity? It can change only the antifouling properties.

Line 172: electric instead of “electronic”

Line 173: hydrophobic what? Additionally, see comment for line 170.

Line 182 to 186: UV-Vis investigations give no useful information on the properties of the layer on the electrodes. So, why not omit? In case of not, please explain, how pyridine can chemically be formed only due to cross-linking.

Line 193: not the addition of GA but the cross-linking reaction

Line 194: pore formation cannot show increase in height, only decrease in valleys. So please explain more correct.

Line 197: why authors showed layers on mice (b) and (c) and on electrodes (d)? Only the latter gives useful information on the properties of the layer for biosensing purposes. So, why not omit?

Line 213: the first sentence is nonsense, please omit or explain correct.

Line 215: Rct subscriped

Line 216: Rs subscriped

Line 217: the charge transfer resistance Rct, which was used for further evaluation. instead of “impedance values”

Line 218: the part from here to line 226 is nonsense. At the end, the fully modified layers on the electrodes were used. So the several steps cannot be compared with each other and give no useful information. So, why not omit?

Line 230: the results show only stability, not directly antifouling properties! The authors should show comparative results (negative control) with blank electrodes under same conditions and the resulting fouling effects.

Line 235: figures are too small. Please enlarge.

Line 240: H2O2 subscripted

Line 251: pH value.

Line 253: potential). Without “)”

Line 262: figure 5 in supplemental is missing

Line 265: why sensitivity? It is only low detection limit!

Line 267: anti-Hb1Ac-BSA/MWCNTs/GA layer (d) does not show more rapid electron transfer compared to anti-HbA1c layer (e). Please explain, why in the opposite was postulated in the main text.

Line 270: 6%HbA1c instead of “%HbA1c”

Line 271: HbA1c instead of “%HbA1c”

Line 272: HbA1c instead of “%HbA1c”

Supplement figure S1: (a) the numbers in the plot must be explained.

Supplement: figure 4 there are errors in the legend

Some literature is missing, e.g.: DOI: 10.1016/j.aca.2019.05.011; DOI: 10.3390/mi11090814,

From line 274 to the end all results and conclusion have to be overworked and must be presented much more clearer.

I am not willing to proceed reviewing without a completely new paper.

Author Response

(The authors gave the same response as above.)

Round 2

Reviewer 1 Report

The authors addressed most of the issues that were risen during the first review round. 

Author Response

The reviewer did not provide detailed comments for the revision of the manuscript. We have checked the revised manuscript for language and style once more time.

Reviewer 3 Report

Line 46-47: there exist commercial tests for HbA1c for Diabetes. So it seem not to be a problem to detect it. Detection itself is independent from formation of HbA1c!

Line 66: limit of detection is influenced by the receptor, not by electrode conductivity, which has no effect. If the authors think yes, please explain!

Line 194-195: UV spectra give information on proteins in the liquid phase, that is right. But the authors have an immobilized protein with other structures than the protein in liquid. So UV gives no useful information for the construction of the biosensor itself.

Line 208: why the authors showed the AFM of layers on mice (b) and (c)? It is not important for the SPCE surface.

General suggestion: the authors should concentrate either on CV or on EIS and not on both and in no case on UV and mice.

Author Response

Great thanks for the reviewer's meticulous and meaningful comments. We have provided point-by-point responses.
